# Microevolutionary dynamics show tropical valleys are deeper for montane birds of the Atlantic Forest

Gregory Thom [1,2 ✉], Marcelo Gehara[3,4], Brian Tilston Smith[1], Cristina Y. Miyaki[2,6] & Fábio Raposo do Amaral[5,6]

Tropical mountains hold more biodiversity than their temperate counterparts, and this disparity is often associated with the latitudinal climatic gradient. However, distinguishing the impact of latitude versus the background effects of species history and traits is challenging due to the evolutionary distance between tropical and temperate assemblages. Here, we test whether microevolutionary processes are linked to environmental variation across a sharp latitudinal transition in 21 montane birds of the southern Atlantic Forest in Brazil. We find that effective dispersal within populations in the tropical mountains is lower and genomic differentiation is better predicted by the current environmental complexity of the region than within the subtropical populations. The concordant response of multiple co-occurring populations is consistent with spatial climatic variability as a major process driving population differentiation. Our results provide evidence for how a narrow latitudinal gradient can shape microevolutionary processes and contribute to broader scale biodiversity patterns.

[1] Department of Ornithology, American Museum of Natural History, Central Park West at 79th Street, New York, NY 10024, USA. [2] Departamento de Genética e Biologia Evolutiva, Universidade de São Paulo, Rua do Matão, 277, Cidade Universitária, São Paulo, SP 05508-090, Brazil. [3] Sackler Institute for Comparative Genomics, American Museum of Natural History, New York, NY 10024, USA. [4] Department of Earth and Environmental Sciences, Rutgers University Newark, 195 University Ave, Newark, NJ 07102, USA. [5] Departamento de Ecologia e Biologia Evolutiva, Universidade Federal de São Paulo, Rua Prof. Artur Riedel, 275, Jardim Eldorado, Diadema, SP CEP 09972-270, Brazil. [6] These authors jointly supervised this work: Cristina Y. Miyaki, Fábio Raposo do Amaral. ✉email: gthomesilva@amnh.org

The tropics harbor more species than temperate regions[1]. While the overall mechanisms causing this pattern are still debated[2,3], in mountains, the latitudinal disparity in species richness is tightly coupled to the interplay between species ecologies and climate[4]. Montane taxa that inhabit temperate environments tend to have broader thermal tolerance and higher dispersal capacities[5,6]. In contrast, tropical montane taxa are subjected to strong elevational zonation, having narrow thermal niches, increased habitat specialization, and consequently limited dispersal[7,8]. These differences should facilitate geographic isolation, differentiation, and promote higher species accumulation in tropical mountains[4]. The link between seasonal climatic variation, thermal stratification, and dispersal limitation across mountains was initially made by Janzen[7], who proposed that "mountain passes are higher in the tropics" after observing that elevational gradients have greater thermal overlap in temperate regions. Numerous studies support this mechanism, suggesting higher levels of genetic differentiation[9,10], reduced niche breadths[8,11–13], and higher species turnover across elevation gradients in tropical mountains[14]. This substantial support was borne out with comparisons of regions at very different latitudes and highly divergent taxa (e.g., the Andes in Ecuador vs the Rockies in the United States[10]), whereas continuous systems where species are distributed across tropical and subtropical climates are far less explored. It remains unclear at what point in the diversification process the major assumptions associated with Janzen's hypothesis start manifesting, in particular, whether these predictions are observable between populations within species. Determining how the initial stages of diversification contribute to the latitudinal diversity gradient in montane regions could be critical to understanding how global patterns of diversity are generated and maintained over deeper scales.

Temperate versus tropical comparisons are usually discrete observations that reflect the extremes of a continuum. A sampling at this scale makes it challenging to isolate the potential effects of climate variation on the latitudinal diversity gradient. In general, temperate species are younger[15], occur in habitats that were drastically affected by historical climatic oscillations[16], and have a distinct evolutionary origin, belonging to clades that have been more successful at dispersing than tropical lineages[17]. By focusing on a shallow temporal scale it is possible to explicitly test Janzen's mechanism while controlling for these confounding effects of history. If the predictions of Janzen's hypothesis are supported within species, populations should progressively occupy wider elevational ranges and broader environmental space as latitude increases[4]. This pattern should lead to more interconnected populations in higher latitudes when compared to the populations in the tropics that are predicted to occur in narrower elevational bands. A key assumption of this model is that regional differences in climate directly affect elevational distributions. Although topography might differ between sampled localities, which could affect the elevational range of species, populations of higher latitudes should progressively occur at lower elevations and occupy a larger proportion of the elevations available in a given region. The evolutionary consequence of these ecological conditions would be that low elevation, high latitude individuals would be more genetically similar than those with restricted, high elevation distributions at lower latitudes. For instance, we expect higher rates of gene flow and larger effective population sizes for the low elevation, high latitude populations. Alternatively, if population differentiation is not associated with latitudinal climatic variability, differences between tropical and temperate montane lineages may only be observable at macroevolutionary scales where taxa occur in contrasting latitudes and have substantially diverged in their traits and histories.

Here, we test whether intraspecific genetic structuring varies across a narrow latitudinal transition, and explore how current and historical environmental variation shapes microevolutionary processes across the gradient. We examine a community of montane birds distributed across the southern Atlantic Forest sky islands in Brazil, a region characterized by two main mountain massifs, the Serra da Mantiqueira and the northern portion of the Serra do Mar, and the Serra Geral and southern portion of the Serra do Mar. These two mountain regions are separated by a low elevation gap of ~250 km wide, overlapping with the formal geographic delimiter of the southern tropical and subtropical regions, the Tropic of Capricorn (Fig. 1). First, we quantify the habitats and elevation occupied by populations of 21 species of birds (Fig. 2) and test for associations between latitudinal variation in climate and elevational distribution. Second, we explore contemporary and historical environmental variation to estimate the relative contribution of these factors in driving local patterns of genome-wide genetic differentiation and historical demography, while accounting for species traits.

This study provides a community-wide assessment of how climatic variability affects population differentiation and habitat occupation across a continuous latitudinal gradient. Our approach allows us to isolate the effects of historical and current habitat configuration and species biology on the genetic diversity of a heterogeneous group of birds to suggest that ecological constraints in the tropics cause greater population differentiation at the onset of speciation.

## Results and discussion

**Climate variability predicts the elevational distribution.** We found that annual temperature variation progressively increases with latitude across the southern Atlantic Forest, leading to a greater overlap of annual temperature ranges across elevation at higher latitudes (Fig. 1). The studied bird species in this region occupy lower elevations as minimum temperatures decrease towards the south, supporting the idea that latitude predicts local patterns of habitat occupancy (Fig. 1). The subtropical populations that occur in lower elevations are more widespread and occupy broader environmental space than the populations in the tropical mountains that are restricted to higher elevations (Fig. 3). For example, in the tropical region, most of the species are exclusively found in *Araucaria* (pine), cloud and montane ombrophilous forests above 900 m, while in the subtropical region species are typically found in a multitude of environments, including the relatively low elevation (c.a., 400 m) semi-deciduous forests in the interior of the continent and the lowland habitats in the southernmost part of their distribution. Variation in the elevational distributions of species across the latitudinal gradients is predicted to be due to climate variability under Janzen's hypothesis. However, in the Atlantic Forest mountains, it could also be caused by subtropical mountains having lower elevational profiles (Fig. 1). We built mixed-effects models to tease apart the effect of both of these factors, climate (annual temperature range) and local elevation (the elevation range available for the species to occur within a given latitudinal degree), on the topographic distribution of species. Our results support that both climate and topography were significant predictors of the elevational range of the species (full model Akaike Information Criterion weight [AIC weight] >0.999; $R^2 = 0.78$; Supplementary Tables 1 and 2) but were impacting the opposing extremes of the species distribution. While the upper elevation limit of the species was predicted by the local elevation (AIC weight = 0.803; $R^2 = 0.91$), indicating that species could occur at higher elevations if they were available, the lower elevation limit was predicted by temperature seasonality (AIC weight = 0.756; $R^2 = 0.87$). Wider elevational ranges

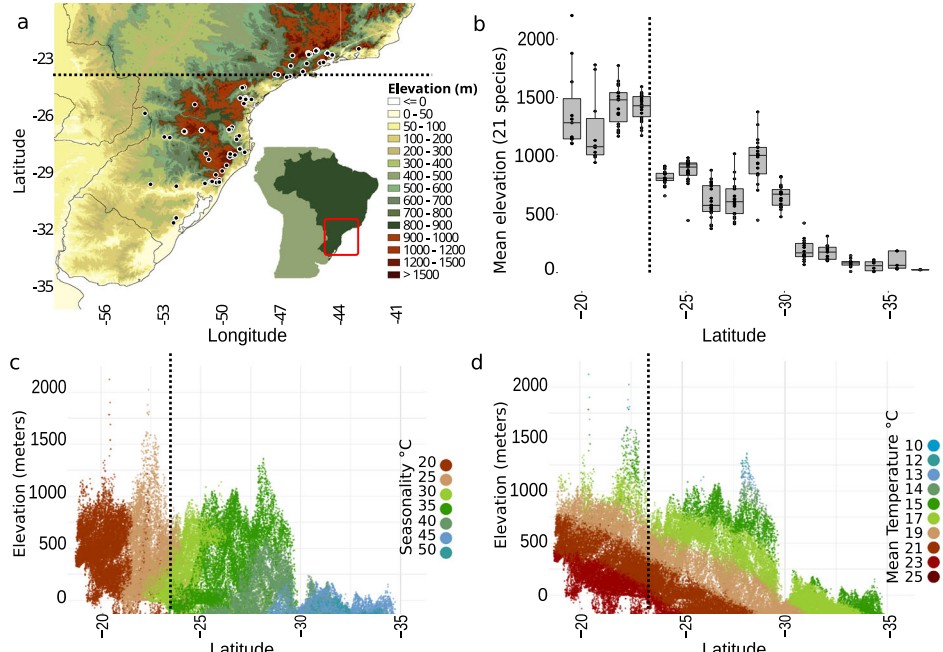

**Fig. 1 Progressively lower elevational distributions of birds follow higher seasonality and lower mean temperatures towards the higher latitudes of the Atlantic Forest sky islands. a** Sampling localities (black dots) in southern Atlantic Forest of 21 species of birds showing the elevation profile of the region in meters (m; Geospatial Information Authority of Japan); **b** Boxplot for the elevation distribution of 21 species of birds showing the community-wide progressive occupation of lower elevations in higher latitudes ($n = 13,986$ occurrence records average across species; mean = 608.1 and SD = 371.3 per species). Source data is provided in Supplementary Data 3. The boxplot is composed by the median (center), the first and third quartiles (lower and upper hinges), and the 1.5 * IQR from the hinge (where IQR is the interquartile range; upper and lower whiskers); **c** Atlantic Forest elevational profile based on the minimum convex polygon buffered by half latitudinal degree around the occurrence sites, showing the progressive increase in seasonality (annual range in temperature) towards higher latitudes; **d** Atlantic Forest elevational profile showing the elevational distribution of mean temperature values across latitude. The dashed line represents the Tropic of Capricorn and indicates the transition between tropical (latitudes <23.5°) and subtropical (latitudes >23.5°) mountain regions.

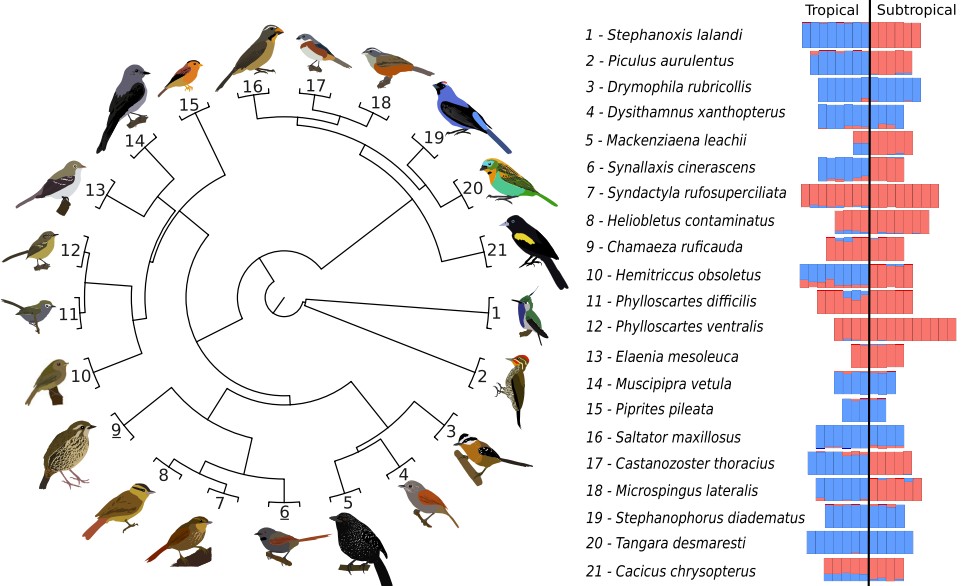

**Fig. 2 Phylogenetic relationship and genetic structure of the focal species of birds that occur in the Atlantic Forest Sky islands (left).** Phylogeny was estimated by ref. [57]. Each terminal was duplicated to represent the tropical and subtropical populations of each species. Bar plots (right) show the genetic structure within each species, where each bar is one individual, and colors represent assigned ancestral populations. Values were estimated using a spatial explicit algorithm that controls for the effects of isolation-by-distance (see Material and Methods).

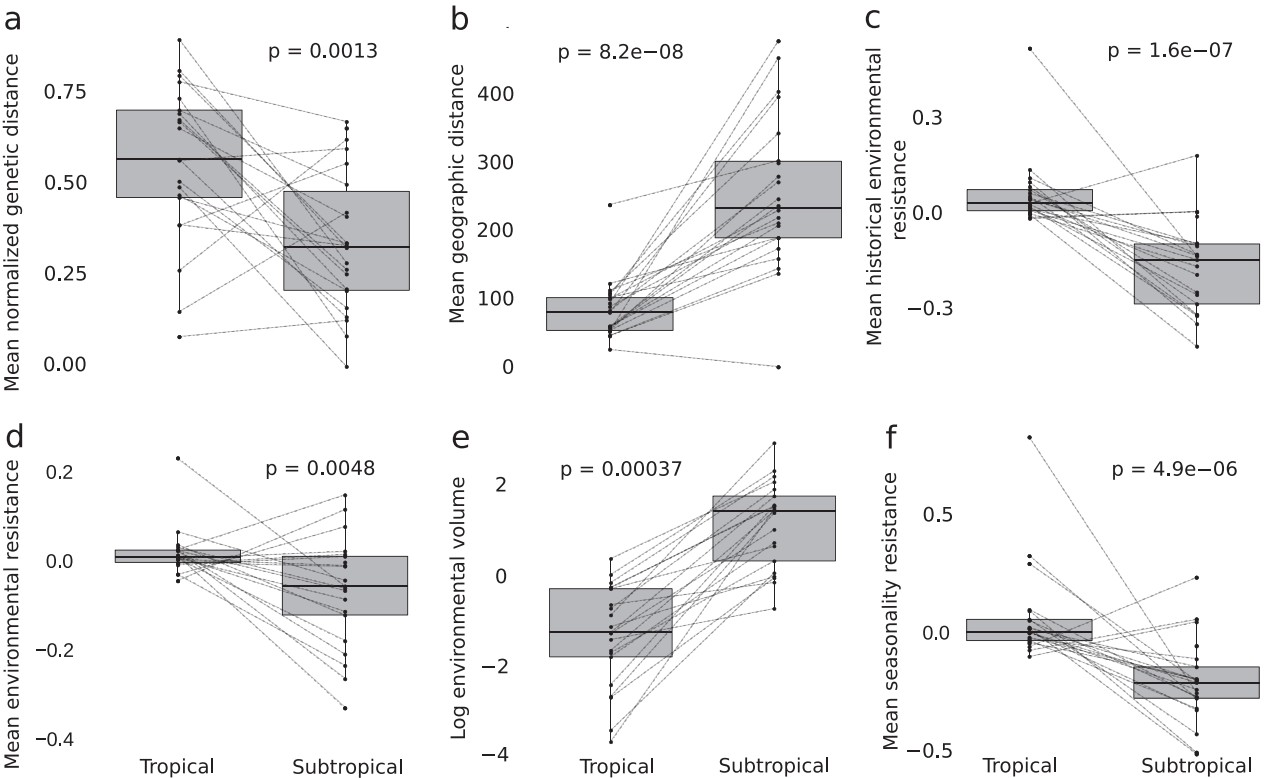

**Fig. 3 Despite being geographic closer, there was higher genetic differentiation and environmental resistance between individuals in the tropical than subtropical populations.** Mean (and SD) genetic and environmental distances for populations of 21 species of birds restricted to tropical and subtropical mountain ranges of the Atlantic Forest. **a** mean (and SD) normalized Nei's genetic distance (n = 17 species pairs); **b** mean (and SD) euclidean geographic distance in meters (n = 17 species pairs); **c** mean (and SD) normalized historical environmental resistance (n = 17 species pairs); **d** mean (and SD) residual of a linear regression between environmental resistance (obtained with a least-cost distance algorithm) and geographic distance (n = 17 species pairs); **e** log environmental multidimensional volume (and SD), based on 19 CHELSA bioclimatic variables (n = 17 species pairs); **f** mean (and SD) normalized seasonality resistance (n = 17 species pairs). p = P value obtained with a two-sided nonparametric Wilcoxon test. The boxplots are composed of the median (center), the first and third quartiles (lower and upper hinges), and the 1.5 * IQR from the hinge (where IQR is the interquartile range; upper and lower whiskers). Lines connect populations of the same species. Source data is provided in the Source Data file.

towards the more thermally seasonal latitudes might affect how individuals disperse across topographically complex terrains, affecting their level of differentiation[18].

**Genetic differentiation supports Janzen's hypothesis**. We showed that a narrow latitudinal gradient (~10°) had a strong effect on the occupied environments and individuals' connectivity. This pattern leads to contrasting levels of genetic differentiation between populations occurring in mountains with distinct climates. We genotyped thousands of SNPs using ddRAD for 21 species (N = 227 individuals) and generated species distribution models (SDMs) for current and past climates. To test the effect of geographic distance (Euclidian), environmental resistance, seasonality, and historical environmental resistance on genetic differentiation, we used mixed-effect models as implemented in ResistanceGA[19]. The most frequent best-fit model AIC weight >0.999 in ten species) included seasonality, environmental resistance, and historical environmental resistance (Supplementary Fig. 1 and Supplementary Table 3). When the tropical and subtropical regions were analyzed separately, our data suggested different patterns of association between geographic and genetic distances. A linear model between normalized genetic distance including all species as a function of geographic distance (DF = 1079; t value = 17.07; p value = <0.0001; R² = 0.3242; Fig. 4) revealed greater genetic differentiation within tropical mountains

despite shorter geographic distances between individuals (Figs. 3 and 4). Therefore, genetic differentiation within populations in tropical and subtropical mountains was likely driven by how much individuals can disperse across valley passes.

We found strong evidence that populations are differently affected by the climate configuration of the Atlantic Forest, with more widespread and interconnected populations in higher latitudes. To test whether intrapopulation gene flow varies between tropical and subtropical regions, we implemented a mixed-effect model with a correlation structure designed to account for multiple species. This approach identified the best environmental predictors of genetic differentiation for all species in each mountain region. We used the optimized resistance matrices obtained in ResistanceGA as predictor variables. Species ecology and time since the colonization of a given region are known to impact genetic differentiation[20]. To control for these effects, we included a morphological surrogate of dispersal ability (Kipp's Index[21]), habitat preference (the major environments occupied by the species[22]), and the minimum time that the species occurred in both regions (represented by the divergence time between populations obtained with our demographic parameter estimation approach) as covariables. For the tropical mountains, the best-fit model contained environmental resistance as a significant predictor of the genetic distance between individuals (Full model AIC = −327.72; AIC weight >0.999; Table 1 and Supplementary Table 3), whereas in the subtropical

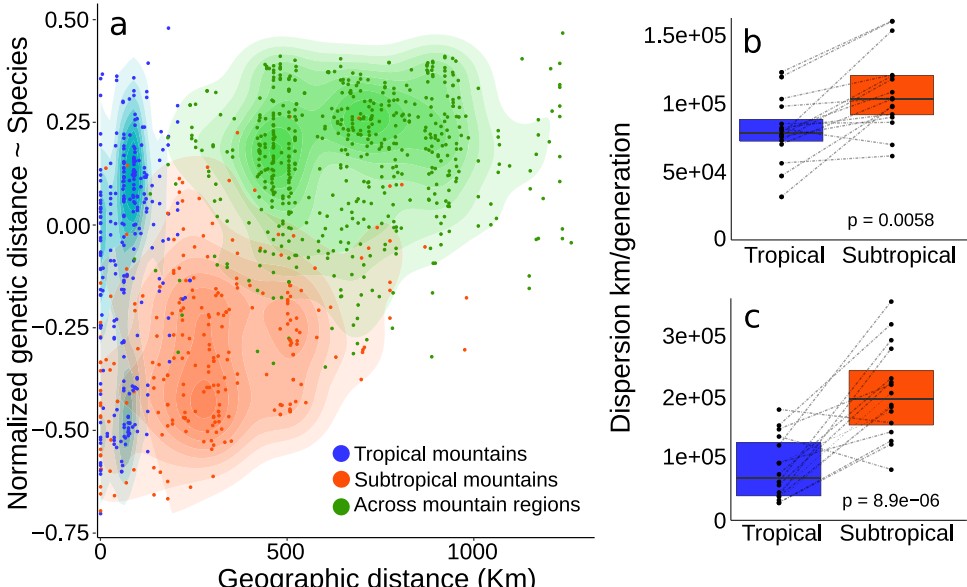

**Fig. 4 Patterns of association between genetic and geographic distances among tropical and subtropical populations. a** Plot of the residuals of a linear model between the normalized Euclidean genetic distance and species, as a function of geographic distance for 21 species of birds restricted to the Atlantic Forest sky islands (regression line: *p* value <0.0001; $R^2 = 0.32$). Colors represent pairwise comparisons within and between mountain regions and color gradients represent the density of points. Geographic distance in kilometers (Km) Effective dispersal in meters per generation for populations occurring in the tropical and subtropical mountains of the Atlantic Forest obtained with the Bidimensional Stepping Stone Model (BSSM); **b** Migrations between localities were scaled by the linear spatial distances ($n = 16$ species pairs); **c** Migrations were not scaled by distances (see Methods for more details) ($n = 16$ species pairs). The boxplots are composed of the median (center), the first and third quartiles (lower and upper hinges), and the 1.5 * IQR from the hinge (where IQR is the interquartile range; upper and lower whiskers). Lines connect populations of the same species. *p* = P value obtained with a two-sided nonparametric Wilcoxon test. Source data are provided in Supplementary Tables 5 and 6.

**Table 1 Support for alternative models of connectivity across the Atlantic Forest mountains from Maximum Likelihood population effect analyses.**

|  | All localities | All localities | Tropical mountains | Tropical mountains | Subtropical mountains | Subtropical mountains |
|---|---|---|---|---|---|---|
| Model | AICc | weight | AICc | weight | AICc | weight |
| IBD | −1043.40 | <0.00 | −309.35 | <0.00 | −327.28 | 0.59 |
| Resistance | −1369.59 | <0.00 | −309.48 | <0.00 | −319.82 | 0.01 |
| Seasonality | −1039.19 | <0.00 | −299.94 | <0.00 | −322.70 | 0.06 |
| Historical resistance | −1380.33 | <0.00 | −302.98 | <0.00 | −321.85 | 0.04 |
| Full_model | −1467.77 | >0.99 | −327.72 | >0.99 | −325.87 | 0.29 |

*IBD* isolation-by-distance, *Full_model* model including the effect of all four resistance matrices.

mountains the best-fit model included only geographic distance (IBD model AIC = −327.28; AIC weight = 0.59; Table 1 and Supplementary Table 3; *p* value <0.001). In all tested cases, divergence time, Kipp's Index, and occupied environment were not significant variables (Supplementary Table 4). In line with Janzen's hypothesis, our results show that tropical populations were more structured in space than the more interconnected subtropical populations. Although our landscape genetics approach showed greater environmental resistance affecting tropical populations, it did not directly quantify the extent individuals had moved across the landscape.

Janzen's hypothesis states that the more restricted elevational distributions in tropical taxa should decrease dispersal and lead to greater genetic differentiation. To test for differences in the effective dispersal of populations in tropical and subtropical mountains, we developed a bidimensional stepping stone coalescent modeling (BSSM) approach that allowed us to simulate data under distinct migration regimes. In the first model, migrations were scaled by the linear geographic distance between sampled localities (IBD-BSSM), while in the second model, a single value was assigned to all

migration parameters (Island-BSSM). We then used a neural network algorithm to explicitly estimate the effective dispersal (De = Nm * geographic distance; where Nm is the product of the effective population number and rate of migration among populations) in each mountain region per species. Under both models, our results support a significantly lower effective dispersal for the populations in the tropical mountains (mean IBD-BSSM = 84.064 km/generation; mean Island-BSSM = 92.849 km/generation) than for populations of the subtropical mountains (mean IBD-BSSM = 112.988 km/generation; mean Island-BSSM = 215.985 km/generation; Supplementary Tables 5, 6 and Fig. 4), supporting Janzen's predictions. The high correlation values between pseudo observed and estimated parameters had high accuracy in estimating both De and Ne (average R2 per species, Ne = 0.89; De = 0.87; Supplementary Table 7). The estimated dispersal distances, along with our mixed-effect modeling, strongly indicate that there was more friction against individuals' movement in the patchy and high elevation tropical environments in comparison to individuals in the subtropical region. The significant differences in dispersal distances and predictors of genetic

**Table 2 Multivariate phylogenetic generalized least squares analyses including the effects of the region (tropical vs subtropical), Kipp's index, and divergence times between populations in distinct mountain regions (Tdiv).**

| Response variables | Region T value | Region P value | Kipp's T value | Kipp's P value | Tdiv T value | Tdiv P value | Full model P value | Full model $R^2$ adj |
|---|---|---|---|---|---|---|---|---|
| Seg. sites | −1.085 | 0.285 | 2.072 | 0.046 | −3.452 | 0.002 | 0.005 | 0.248 |
| Pi | −1.527 | 0.136 | 0.993 | 0.327 | −3.190 | 0.003 | 0.012 | 0.208 |
| Gen. distance | −3.609 | 0.000 | −1.356 | 0.184 | −0.385 | 0.702 | 0.003 | 0.251 |
| Tajima's D | 2.319 | 0.027 | −1.667 | 0.105 | 2.336 | 0.026 | 0.013 | 0.203 |
| Ne | −1.642 | 0.110 | 1.058 | 0.297 | −0.519 | 0.607 | 0.291 | 0.024 |
| Pop. siz. change | 2.438 | 0.051 | 0.204 | 0.839 | 0.794 | 0.432 | 0.101 | 0.092 |
| τ | 1.496 | 0.144 | 0.053 | 0.958 | 3.468 | 0.001 | 0.006 | 0.245 |
| Env. vol. | 3.431 | 0.002 | −0.035 | 0.972 | 0.473 | 0.640 | 0.015 | 0.196 |
| Geo. distance | 8.277 | 0.000 | −0.517 | 0.609 | −0.638 | 0.527 | 0.000 | 0.642 |
| Res. residuals | −2.271 | 0.030 | −0.843 | 0.405 | 0.657 | 0.515 | 0.128 | 0.077 |
| Saz. residuals | −3.390 | 0.002 | 0.352 | 0.727 | −1.027 | 0.312 | 0.013 | 0.205 |
| Sta. residuals | −4.653 | 0.000 | −1.828 | 0.076 | −0.174 | 0.863 | 0.000 | 0.377 |

*Seg. sites* average number of segregating sites, *Pi* average nucleotide diversity, *Gen. distance* normalized average pairwise genetic distance between individuals, *Ne* effective population size, *Pop. siz. Change* the proportion of the ancestral population size compared to the current size, *τ* time since population size change, *Env. vol.* multidimensional environment space, *Geo. distance* average geographic distance between individuals, *Res. residuals* an average of the residuals of linear regression with environmental resistance in the function of geographic distance between individuals, *Saz. residuals* an average of the residuals of linear regression with seasonality resistance in the function of geographic distance between individuals, *Sta. residuals* an average of the residuals of linear regression with the historical environmental resistance in the function of geographic distance between individuals.

differentiation among groups were likely a product of the current climate configuration, where seasonality increases towards the subtropical region, but these patterns could also have been influenced by historical population dynamics.

**Historical demography was not associated with mountain regions.** Changes in historical population sizes could shape the observed patterns of pairwise genetic distance within the tropical and subtropical populations. For example, more intense climatic oscillations affecting higher latitudes could lead to population bottlenecks reducing the average genetic distance among individuals. Similarly, more unstable environments in the subtropics might lead to higher rates of population extinction, followed by recolonization and founder effects from tropical populations[15,23]. We tested whether mountain regions could predict genetic summary statistics and demographic parameters by implementing Phylogenetic Generalized Least Squares (PGLS) models (Table 2, Supplementary Table 8, and Supplementary Fig. 2). We estimated demographic parameters based on a two-population model (tropical and subtropical populations) allowing for gene flow and population size changes over time using Fastsimcoal v. 2.6[24]. To estimate population structure, we assigned individuals to ancestral populations while controlling for geographic distance in conStruct[25] (Fig. 2 and Supplementary Table 9). Mountain regions (tropical or subtropical) significantly predicted average pairwise genetic distance and Tajima's D but failed to predict the size, magnitude, and timing of shifts of Ne (effective population size) (Table 2). Interpreting the significant Tajima's D values was confounded by the greater population structure in tropical populations, which may have biased the metric towards finding more support for expansion (Tajima's D tropical: mean = −0.315, SD = 0.169; Tajima's D subtropical: mean = −0.234, SD = 0.129). The relatively stable population sizes over time in each mountain range do not support a scenario with local extinction in the subtropics followed by recolonization out of the tropics[23]. Despite the expectation that more seasonal mountains in the subtropical region should harbor more unstable populations, our results support the southern Atlantic Forest overall as a relatively stable region for cold-associated species as reported in previous studies[26,27].

**Signal versus noise: the benefits of community-wide sampling.** We performed a comprehensive set of analyses on a multi-taxon genomic dataset that allowed us to quantify fine-scale patterns of genetic structure to test Janzen's hypothesis within species. Despite the relatively reduced number of samples per species (mean = 11.2), our results remained consistent among species and between independent approaches. We analyzed our dataset using three alternative methods by (1) looking at individual species considering their whole distribution within the Atlantic Forest, (2) combining all the species in a single analysis testing environmental predictors for tropical and subtropical regions, and (3) developing a coalescent-based method that allowed us to explicitly estimate the effective dispersal of individuals between sampled localities. By analyzing individual taxa we were able to explore the variation in the main predictors among species, and by combining many species in the same analyses we reduced the effect of low sampling for individual taxa, capturing the major process driving community-wide genetic differentiation across a latitudinal gradient. In addition, our coalescent-based method was an accurate way to assess how effective dispersal varies between populations using a reduced number of samples.

Analyzing localities at different geographic scales or with uneven sample sizes could lead to biased results. For example, sampling at a closer geographic distance within the subtropical region than the tropical region could artificially produce lower estimates of genetic differentiation in the former. However, we used a similar number of samples in both regions (Fig. 2 and Supplementary Data 1). Additionally, our sampling scheme in the subtropical region was more widespread than in the tropical region (Supplementary Figs. 3, 4). Although more nuanced sampling biases may impact the precision of our inferences, our results are likely conservative in characterizing the higher degree of genetic differentiation within populations occurring in the tropical mountains.

**The latitudinal gradient in montane species starts at the microevolutionary scale.** Species distributed across latitudinal gradients often display variation in physiological traits, with higher-latitude populations being more plastic and having wider thermal tolerances[28–30]. Although we cannot rule out that Atlantic Forest birds are locally adapted to distinct climates, our results likely suggest that phenotypic plasticity and elevational compensation following climatic variation can lead to similar patterns to the ones observed in studies comparing highly divergent organisms with contrasting thermal tolerances[10]. Biotic

interactions could also limit the elevational distributions of montane species[31]. Little is known about interactions between species of birds in the Atlantic Forest[32], but interspecific aggression is a pervasive behavior in birds[33], and competitive interaction likely drive elevational divergence[34]. Testing for both, spatial variation in physiological traits and biotic interactions could produce stronger support for the effects of climate variability driving local patterns of gene flow. However, the presence of a strong climatic gradient and the concordant response of multiple co-occurring populations are consistent with spatial climatic variability as a major factor driving greater population differentiation in tropical than subtropical populations (but see ref. [35]), confirming Janzen's predictions within species.

Our results show higher differentiation in tropical mountain populations, consistent with the view that the historical stability and strong elevational zonation promotes higher rates of geographic isolation and subsequent speciation[4,8], which runs counter to the emerging idea that diversification was faster in temperate regions[36–38]. However, we did not directly estimate rates of differentiation, and not accounting for time in the landscape, beyond population split times, could bias interpretations. At broader geographic scales, the degree of phylogeographic structuring in birds was previously shown to be best predicted by species age[15], which applied to this system could indicate tropical populations have greater structure, in part, because they have been in lower latitudes longer. Despite the possibility of differential time in the landscape, genetic differentiation within the Atlantic Forest tropical mountains was likely formed after the divergence between most tropical and subtropical populations, following the upslope movement of montane environments associated with past climatic cycles[39]. Future studies that model the deeper biogeographic history of these montane taxa will be able to clarify the role of time-dependency in interpreting rates of genetic structuring across the latitudinal gradient.

The contribution of the microevolutionary processes reported in this study to macroevolutionary patterns is unclear, but our results show a substantial disparity in the initial stages of speciation between tropical and subtropical regions. The greater differentiation within tropical populations might not lead to higher species formation, which could be limited by time and diversity-dependent processes[37,40], such as ecological competition[41]. Recent work has supported an association between differentiation and species formation, with higher predictability of population structuring and speciation in the tropics[42], indicating a link between micro and macroevolution to some degree. Regardless of the evolutionary outcomes of this genetic structuring, our results suggest that mountains of southeastern Brazil, a small part of the biodiverse rich Atlantic Forest, have an ideal geographic configuration to explore the effects of climate in population differentiation, and harbor multiple isolated populations, some of which may be cryptic species awaiting discovery and protection.

## Methods

**Genomic sampling**. A total of 227 tissue samples from 21 species were obtained in the field or requested from natural history collections (mean = 10.8, SD = 3.08 samples per species; Supplementary Data 1). Specimens were collected under the ICMBio/SISBIO permits 30835–3, 43297–1, and 47205–1 and approved by the Animal Ethics Committee of the Universidade Federal de São Paulo (CEP 0069/12, CEUA 3139240719, and CEUA 6433170817). All genetic samples were included on the SisGen platform under the protocols R2F97C8 and RB9572E. Total DNA was digested with two endonucleases (*Pst*I and *Msp*I) to produce a reduced genome library that was sequenced in an Illumina platform by the University of Wisconsin Biotechnology Center (UWBC, Madison, WI). Sample demultiplexing, de novo assembly, and SNP calling for each species was performed in ipyrad v0.9.18[43], using the same set of parameters for all species. To minimize biases that can arise in comparative RADseq datasets due to variation in genetic diversity among species[44], we used relatively loose filtering for the clustering threshold (0.90

clust_threshold). This setting allowed for a 10% sequence divergence between reads, which reduces the impact of allele dropout on species with more genetic diversity. Additional post-sequencing filters were not implemented because they have marginal impacts on estimates of genetic differentiation, admixture, and demographic parameters[45]. Only loci with less than 25% missing information were retained. To reduce the possibility of linked SNPs, one variant per locus was randomly selected for downstream analyses. A detailed description of our bioinformatics pipeline is available in the "Supplementary Information" (Methods full description).

**Genetic structure between and within mountain regions**. We tested for the number of ancestral populations (K) for each species complex and clustered individuals to populations, controlling for the effects of geographic distance, using conStruct[25] in R v3.6 (R Core Team, 2019). ConStruct simultaneously estimates continuous and discrete patterns of population structure by assuming a rate of decay in the relatedness among individuals as a function of the geographic distance. Thus, discrete groups are only assumed when genetic variation significantly deviates from isolation-by-distance. We used conStruct to explore the presence of genetic structure between the two main mountain blocks of the Southern Atlantic Forest by testing if a model with two layers (K) could better explain genetic variation than a model including a single layer. For each conStruct run, we performed 50,000 iterations discarding the first 50% as burn-in. We assumed layers with a relative contribution >5% to the total covariance of the model were significant. To obtain pairwise genetic distance matrices among individuals across the entire distribution of the species, we estimated Nei's genetic distance[46] with Adegenet v1.3[47] in R v3.6 using SNP matrices.

**Estimation of historical demographic parameters**. To estimate historical demographic parameters, we used Fastsimcoal v2.6[24], which uses the site frequency spectrum (SFSs) as a summary statistic to approximate the likelihood of a demographic model given the data. We used easySFS v1.0 (https://github.com/isaacovercast/easySFS; access date: 05/01/2020) to estimate the projection of SNPs that maximize the number of segregating sites in the Joint SFSs (considering two populations) for each species. To estimate demographic parameters, we built a model with two populations (tropical and subtropical) allowing for an instantaneous demographic size change after divergence (expansion or contraction) in each population, and constant gene flow between populations. Search intervals for the approximate likelihood computations are available in Supplementary Data 2. For each species, we performed the composite likelihood search using 50 independent runs with a maximum of 50 cycles of Brent's algorithm and 100,000 genealogical simulations per generation. Under this model, we were able to estimate the amount of gene flow and the effective population sizes ($N_e$) as well as the time and the magnitude of population size changes. We obtained a confidence interval for the estimated parameters with 50 parametric bootstrap replicates. Each replicate involved selecting the parameters of the run with maximum likelihood, simulating a new SFS under these parameters, and re-estimating parameters under these newly simulated SFS. This approach allowed us to indirectly observe the fit of the model to the data by checking if the parameters estimated from the observed SFS fell within the range obtained from the bootstrap values. Lastly, to estimate the number of segregating sites (S), nucleotide diversity ($\pi$), Tajima's D per population, and pairwise $F_{ST}$ between populations, we used PipeMaster v0.0.9[48] in R. Species with less than three individuals per population were excluded from the analyses.

**Species distribution models and occupied environmental space**. To obtain species occurrences, we data mined and filtered records from the Global Biodiversity Information Facility, GBIF.org (04/28/2020; GBIF Occurrence Download https://doi.org/10.15468/dl.8383sp). After filtering, our final dataset had 13,986 occurrence records (mean = 608.1 and SD = 371.3 per species; Supplementary Data 3). We reduced sampling biases and homogenized the density of occurrences across space, by applying a spatial thinning using a 10 km distance in spThin v0.1.0.1. The extent of current and past climatic suitable areas across the southern Atlantic Forest for our focal taxa was accessed through species distribution models (SDMs). We used the 19 bioclimatic variables from the CHELSA database[49] with a spatial resolution of 30 s. Current and past habitat suitability was estimated in Maxent v3.4.1[50] (Supplementary Data 4). To avoid model overfitting, we evaluated distinct feature classes and regularization multipliers using ENMeval v0.3.0. To remove extrapolation from the projected SDMs generated by combinations of climatic variables not represented by the training dataset, we used a multivariate environmental similarity surface (MESS) analysis in dismo v1.3-3. The occupied environments by tropical and subtropical populations were estimated with a multidimensional ordination of the climatic variables. Values of the CHELSA bioclimatic variables at 30 s resolution were extracted from occurrence localities in raster v2.6-7. Variables were normalized to the mean and submitted to a PCA as implemented in prcomp v3.6.1 in R v3.6. Scores of the first four components, representing on average 96.6% of the total variation (average across all species), were used to compute a per-species multidimensional volume of the occupied climatic space using hypervolume v2.0.12[51] in R. A detailed description of our SDMs is available in the "Supplementary Information" (Methods full description).

To test for the predictors of species' elevational distribution across latitude, we built mixed-effect models exploring the relative contribution of temperature seasonality and local elevation using lme4[52] in R. This approach was used to test if the decrease in total elevation of the Atlantic Forest mountains towards higher latitudes (Fig. 1) could predict the elevational distribution of the species (Supplementary Fig. 5), rejecting the role of climate variability. We calculated the maximum available elevation and mean seasonality for each latitudinal degree within a minimum convex polygon around our filtered species occurrence data. For each species with more than 30 records for a given latitudinal degree, we estimated four response variables: (1) the elevational range, (2) minimum elevation, (3) maximum elevation, and (4) the proportion of the available elevations that the species occupy. We removed the top and bottom 5% of the occurrence records based on elevation to avoid the inclusion of outliers records outside the elevational distribution of the species. For each of the four models described above, we performed model ranking using AIC weights on three models: (1) full model, including the interaction of both, seasonality and local elevation, (2) just seasonality, and (3) just local elevation. Species-level was included as a random effect in the models.

**Predictors for the spatial variation in population differentiation**. Based on the environmental analyses, our approach produced four variables that capture different mechanisms potentially affecting individuals' connectivity, namely the geographic distance, environmental resistance, temperature seasonality resistance, and historical environmental resistance. The environmental resistance was derived from the current climate projections and reflects the degree of resistance for individuals to move in space according to the suitability values of SDMs. The seasonality resistance was derived from the fourth bioclimatic variable of the CHELSA database (Bio4 = Temperature Seasonality). In this variable, higher values of seasonality are translated into less resistance among individuals. Lastly, the historical environmental resistance matrix was generated by averaging the binary presence-absence layers of each projected time slice: (1) Pleistocene: Last Interglacial (ca. 130 ka), (2) Pleistocene: Last Glacial Maximum (ca. 21 ka), (3) mid-Holocene: Northgrippian (8.326–4.2 ka), and (4) late-Holocene: Meghalayan (4.2–0.3 ka). Landscape resistance for each variable was obtained by calculating the cost distance along the least-cost path between samples using gdistance v1.3. We assessed the individual and the combined effects of landscape resistance layers on genetic distance using ResistanceGA v4.0[19] in R. Nei's genetic distance was used as the response variable and the scaled and transformed landscape layers were used as predictor variables including all possible combinations of resistance matrices. All species were sampled for at least five localities. To test for a community-wide predictor of genetic differentiation, we implemented a maximum likelihood population effect model (MLPE) in nlme v3.1[53] and corMLPE v0.0.2 (https://github.com/nspope/corMLPE) in R. This approach fits a mixed-effect model with a correlation structure designed to account for the nonindependence of pairwise distance matrices, treating the residuals of the pairwise comparisons as the sum of two random populational level effects[53]. We normalized the response and predictor variables, genetic distance, and environmental distances, respectively. To test tropical and subtropical mountains' specific drivers of genetic differentiation, we fitted a model including all individuals for all species as well as models for each mountain region independently. We performed model ranking using AIC weights. A detailed description of our landscape genetics approach is available in the "Supplementary Information" (Methods full description).

**Bidimensional stepping stone coalescent modeling**. To simulate the effect of distance we built for the tropical and subtropical populations of each species, a bidimensional stepping stone model (BSSM) where each locality represented a population with assigned effective population size (Ne). Populations were connected through migration (Nm) to two of their nearest neighbors according to the linear geographic distance among them. We only included species with at least three sampled localities. We assigned the same Ne to all localities, and values were sampled from a uniform distribution (min: 20,000; max: 400,000). The amount of migration among localities varied. We built two versions of the BSSM which parameterized migration in different ways. In the first version, migrations were scaled by linear geographic distance (IBD-BSSM). To do so, we included an effective dispersal parameter (De), which represents the movement of alleles in space per generation. To calculate the number of migrants, we divided De by the linear distance. For instance, if De = 50 Km and the linear distance between two localities, Ne1 and Ne2, is 100 Km, there will be 0.5 Nm per generation for each migration direction (i.e. Nm1 < 2 and Nm2 < 1). The De values were sampled from a uniform prior distribution ranging from 1–200 Km. It is unrealistic to expect a perfect correlation between migration and linear distance, thus we added a noise parameter, represented by a percentage sampled from a uniform distribution (0–1). The noise was generated by randomly making the scaled migrations higher or lower by that sampled percentage. In the second model (Island-BSSM), a single Nm value was sampled from a uniform distribution (0.25–5) and assigned to all migrations between localities. Each Nm varied according to the sampled noise parameter as above. In this case, De was conditional on the sampled Nm and was calculated by multiplying the average linear distance by Nm. For each region, tropical and subtropical, and each species, we generated the two models and sampled parameters using R 3.6.3 (R Core Team, 2020). We then used the sampled

parameters to simulate the following summary statistics using msABC[54]: number of segregating sites, nucleotide diversity, pairwise $F_{ST}$, pairwise percentage of shared polymorphism, fixed polymorphism and private polymorphism, and Tajima's D. We ensured that the simulated models fitted the observed data by performing a PCA and plotting the first ten PCs of simulated statistics vs observed (Supplementary Data 5). We also generated goodness-of-fit plots using the gfit function of abc v2.1 in R v3.6[55]. We simulated 10,000 datasets per species per region (tropical and subtropical), these were used to train a neural network (NN) regression model, using the R v3.6 interface to Keras v2.3 (https://github.com/rstudio/keras), to estimate the two main parameters of interest: Ne and De. We simulated an additional 1000 datasets for testing the model. We built a NN with two hidden layers with 32 nodes each and an output layer with a single node, all with a relu activation. We used the rmsprop as an optimizer and a Mean Absolute Error (MAE) for loss and accuracy metrics. For each parameter, we trained the NN for 1000 epochs with a validation split of 0.1. For additional evaluation metrics, we plotted and calculated the correlation coefficient of true versus estimated values for the testing data. We performed four replicates of the NN regression and used the median as the final value. We also calculated the standard deviation to evaluate variation among replicates.

**Phylogenetic generalized least squares models**. To test for the association between environmental variables and genomic metrics with mountain regions and control for the relatedness among species, we applied phylogenetic generalized least squares (PGLS) models using caper v1.0.1[56] in R. We ran 12 independent univariate analyses varying the response variables with mountain region as the predictor variable (tropical or subtropical; Table 2). For each analysis, we selected one population genetics summary statistic, demographic parameter, or environmental variable calculated for all populations (two populations per species; individual response variables are described in Table 2). We summarized genetic and environmental distances by averaging pairwise comparisons of individuals occurring in the same mountain region. We also included Kipp's Index and divergence times in every model as additional predictor variables to control for dispersal ability and species ages, respectively. All variables were normalized by square-root transforming right-skewed data. The phylogenetic tree including all species was retrieved from birdtree.org[57]. We downloaded 1000 trees sampled from the posterior distribution of the "Ericson All Species" analyses using the V2.iii backbone calibration and built a maximum clade credibility tree on TreeAnnotator[58]. To test for the phylogenetic signal of the predictor variables, we calculated Blomberg's K[59] with adiv v2.0[60] in R. Values close to 0 indicate lack of a phylogenetic signal in the data while values near 1 represent Brownian character evolution. Significance was assessed with 1000 permutations where variable values were shuffled between species.

**Reporting Summary**. Further information on research design is available in the Nature Research Reporting Summary linked to this article.

## Data availability

All data needed to evaluate the conclusions of this study are present in the manuscript and/or in the "Supplementary Information". Additional data related to this study is available at https://github.com/GregoryThom/ and at https://doi.org/10.5281/zenodo.5510615. The genetic data generated in this study have been deposited at the NCBI Short Read Archive under the BioProject PRJNA723198 at https://www.ncbi.nlm.nih.gov/bioproject/PRJNA723198. The occurrence records used in this study were obtained from the Global Biodiversity Information Facility and are available at https://doi.org/10.15468/dl.8383sp. The maps on Fig. 1a, Supplementary Figs. 3 and 4 were designed using QGIS (https://qgis.org/en/site/), and the Elevation - Global version 1 shapefile obtained at https://globalmaps.github.io/el.html (Geospatial Information Authority of Japan). Climatic data were obtained from https://chelsa-climate.org/downloads/. Additional data generated in this study are provided in the Supplementary Data files. Source data are provided with this paper.

## Code availability

All code needed to replicate this study is available at https://doi.org/10.5281/zenodo.5510615 (https://github.com/GregoryThom/), and https://doi.org/10.5281/zenodo.5512912 (https://github.com/gehara/BiSSM).

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

## Acknowledgements

We would like to thank K. Provost, L. R. Moreira, J. Merwin, V. Chua, E. Tenorio, A. Carnaval, A. Paz, K. Mercier, L. Martins, R. Mascarenhas, C. French, J. Cracraft, P. Sweet, T. Trombone, B. Bird, A. Aleixo, L. Musher, I. Franz, E. Shultz, A. Nuss, V. Piacentini, C. Assis, C. Fontana, A. Morales, F. Burbrink, and G. Seeholzer. We also thank the Laboratório de Genética e Evolução Molecular de Aves—LGEMA, Museu Paraense Emílio Goeldi—MPEG, Museu de Ciências e Tecnologia da Pontifícia Universidade Católica do Rio Grande do Sul—MCT-PUCRS, Museu de Zoologia da Universidade de São Paulo—MZUSP, and Coleção Zoológica da Universidade Regional de Blumenau—FURB for tissue samples. This study was co-funded by FAPESP and the BIOTA program (2011/50143-7, 2011/23155-4, 2013/50297-0, and 2018/03428-5), CNPq/MCTIC research productivity fellowship (FRA, 312697/2018-0; CYM, 303713/2015-1, 306204/2019-3), NASA through the Dimensions of

Biodiversity Program of the National Science Foundation (DOB 1343578 and DOB 1831560). G.T. was granted by FAPESP scholarships (2018/17869-3 and 2017:25720-7) and the Frank M. Chapman memorial fund of the American Museum of Natural History. M.G. postdoctoral fellowship was funded by the Gerstner Family foundation—Gerstner Scholars program of the American Museum of Natural History. B.T.S. was funded by US National Science Foundation award DEB-1655736

## Author contributions

G.T., F.R.d.A. and B.T.S. designed the study. G.T. and F.R.d.A. collected the samples in the field. G.T., B.T.S. and F.R.d.A. wrote the manuscript with comments and suggestions from C.Y.M. and M.G. G.T. and M.G. conducted the bioinformatics and statistical analyses. G.T. designed the bird drawings. These authors jointly supervised this work: F.R.d.A. and C.Y.M.

## Competing interests

The authors declare no competing interests.
