## [Peer Review File · Nature Communications]

Microevolutionary dynamics show tropical valleys are deeper for montane birds of the Atlantic ForestREVIEWER COMMENTS

Reviewer #1 (Remarks to the Author):

Thom et al. examine latitudinal differences in intraspecific population differentiation across space, focusing in on a latitudinal transect across the tropical/subtropical boundary in Brazil. They use well-sampled RAD-Seq datasets from 21 species to test for differences in the environments occupied, genetic structure, and dispersal between the subtropical and tropical regions. They find that species occupy lower elevations in the subtropics than the tropics. Tropical populations show greater differentiation across space and lower dispersal than subtropical counterparts. Tropical differentiation is associated with environmental turnover, whereas any subtropical differentiation is associated with distance alone.

Overall, I thought the conceptual focus on latitudinal differences in montane divergence was of fundamental interest, I appreciated the approach of examining the same species across a latitudinal transect, I thought the sampling was quite good, and I found the presentation to generally be effective. I have a few concerns regarding the data, I think the text could be clearer in places, and I thought some of the methods were lacking in detail or had redundant information provided across the Results, Methods, and Discussion. However, I think these issues could be corrected with a revision of the manuscript and would yield a very nice study of general interest.

Major comments:

1. Comparability of data among species and regions

Are ddRAD datasets comparable across species? If the same parameters are used for assembling datasets across species, those that contain more divergent alleles might be disproportionately affected by e.g., allele dropout. Might this impact any of the reported results? I know there is some prior work on these issues in RAD-Seq data (e.g., Ilut et al. 2014, Harvey et al. 2015) and ddRAD specifically (e.g., Cumer et al. 2021). I would like to see at least a discussion of the comparability issue.

Of greater concern for this study would be a bias in diversity in the tropical versus subtropical regions. Greater sampling in the tropical region might lead to the discovery of more sites that are variable among individuals in that region. It does not look like more individuals were sampled in the Tropics based on

e.g., Figs. 1 and 2, but I cannot be sure without access to Table S12. Greater sampling in one region would not bias genetic distances based on all sites in the data, but it could lead to upward bias in differentiation measures based on variant sites only. Was Nei's genetic distance based on all sites or variants only? Were all sites used for consStruct analyses or just SNPs? Please provide this information in the text.

2. PGLS presentation needs improvement

I understand the utility of the PGLS analysis to identify any drivers of additional demographic differences between the tropics and subtropics. However, I was confused why migration rates were not included as response variables, since the Fastsimcoal model included gene flow. I know the BSSM simulations already identified a difference in migration rates between the two regions, but would be curious to know if PGLS had the power to detect this. If not, it may indicate that PGLS using the parameters from Fastsimcoal lacks sufficient power to detect demographic differences between the regions. Also, the significant Tajima's D results needs explanation (see Minor Comments).

I also found that methodological details were lacking for this PGLS analysis. What were the units of observation? Given that "region" was a predictor variable, each species must have been split into two so there would be separate values for tropical and subtropical populations, correct? This is not clear from the text (either the Results, the Methods, or the SI Methods). How were variables summarized within each population? Were segregating sites and π calculated across all samples from those populations or were they calculated for each individual and a mean obtained? How were distance metrics summarized – were they calculated as pairwise values for the subset of individuals in each subpopulation (tropical and subtropical)? The predictor variables are also unclear. Is region a simple binary variable? How is TDiv calculated for each subpopulation given that there should be only one value for each species. To what does "Full model" refer – a multivariate analysis with region, Kipp's index, and TDiv all as predictors? This information could be added to the SI, which currently contains little more detail than the main text Materials and Methods.

The goal of the second PGLS analysis (testing level of consStruct structure versus environmental space) is less clear to me. I guess it was to identify whether the degree of environmental difference between the tropical and subtropical areas inhabited by a species determined whether it had a phylogeographic break between them? This is perhaps not directly relevant to the major questions, so this analysis might be dropped. If it is retained, a sentence of justification should be added to the Results/Discussion section where the test is first introduced. Also, in the methods, please describe the units of observation (I presume this was each species, unlike the first PGLS analysis) and how the difference between environments of the two regions was calculated.

3. Precision and clarity of text

I found the text unnecessarily wordy or vague in some places. For example, in the Abstract the sentence:

“However, distinguishing the background effects of species history and trait divergence from causal processes driving higher tropical diversity is challenging given the extent of evolutionary time separating taxa along latitudinal gradients.”

might be clarified as:

“However, distinguishing the impact of latitude versus phylogeny is challenging due to the evolutionary distance between tropical and temperate assemblages.”

It may sound drier, but it is more straightforward to interpret. There are other places where the wording can be tightened, some of which are indicated in the minor comments below.

4. Redundant methods details

The same methodological information is often duplicated (sometimes verbatim!) between the main text Materials and Methods, the SI Materials and Methods, and sometimes even in the Results and Discussion (see examples in Minor Comments). At the same time, key methods information is sometimes missing (see PGLS comment above). This meant I had to search multiple places to find key info and sort through the same details in each place, only to fail to find what I was looking for. Please try to reduce redundancy in favor of providing complementary detail in the SI.

Minor Comments:

49: “...and support that valley...” is awkward. Better: “...and support the [idea/theory] that valley...”. There are several other places with the same issue (e.g., lines 129, 143).

73: This idea would be clearer if you replaced “with contrasting” with “at very different”.

75: Replace “while” with “whereas”.

78: Based on these sentences, two knowledge gaps are being explored – (1) “continuous systems” containing close relatives and (2) intraspecific (rather than species-level) diversification. Perhaps make this clearer?

95: “Progressive” should be “progressively”.

116: Missing a word between “factors” and “driving” (“in”?).

119: Remove comma after “configuration”.

125: “predict” should be “predicts”.

128: “on” should be “at”.

128: Which montane birds? The same 21 species used in genetic analyses?

169: What are “valley passes”? Just valleys?

182: It is misleading to say here that the best-fit model supported a significant effect of environmental distance. You could either present a pairwise comparison of otherwise-identical models with and without environmental distance, or you could say “the best-fit model contained environmental resistance variables”.

183: “distance as significant” should be “distance as a significant”.

190: My understanding is that dispersion refers to the pattern of distribution of individuals across the landscape, whereas dispersal refers to movement. It seems the two terms are used interchangeably here? I would suggest using dispersal when movement is involved.

205: "individuals" should be "individual".

226: Was Tajima's D lower in the subtropics? This might indicate population expansion and is worthy of discussion. Maybe there was just not enough power to identify differences in the population sizes from demographic modeling.

234: What was the purpose of this PGLS analysis? To determine what traits of species predicted whether there was a phylogeographic break between the two regions? Is this relevant? What does it mean that current environmental resistance is a significant predictor? Species with more environmental turnover among samples overall are more likely to have a phylogeographic break?

240: This paragraph is odd in that it contains conclusions and then discusses sampling issues, which don't really go together. I would suggest moving the conclusions sentences to the very end of the paper. Also, you might support the statements about sampling issues with quantitative information from the results or with additional sensitivity tests. This would also be a good place to include information addressing Major Comment 1.

249: "taxa" is the plural of "taxon" (see also line 251).

253: Is this accuracy demonstrated with simulations or similar? I could not find these details. If not, remove this sentence.

261: Add "an" before "extensive"?

267: Hyphenate "higher-latitude".

279: Could you more explicitly tie back in to Janzen's hypothesis here? State explicitly if you confirm or refute it?

283: This is a good discussion of the paradox of lower speciation but greater population differentiation in the tropics. My opinion is that the authors are correct that the likely explanation is a greater lag-time for differentiated populations to become species in the tropics. More work is certainly needed on this topic!

298: Insert "may be" between "which" and "cryptic".

303: "were" should be "was".

360: Clarify the normalization procedure. Was this a simple transformation?

650: State what method was used to obtain the barplots.

650: "spatial explicit" should be "spatially explicit".

668: Perhaps clarify that this regression was used to center the values across species?

Table 1: Clarify that full model is the combined resistance layers ("full model" is not mentioned in the text).

Table 2: What are the predictor variables (columns) here? Specifically, does region just refer to a binary variable or is this the environmental difference between regions as indicated in the SI methods? What is the "Full model"

Supplemental Information:

9: Check on table references (S5 does not contain sample information).

111: This exact sentence appears in the main text. Please reduce redundant text.

116: Almost this exact sentence appears in the main text. Please reduce redundant text.

118: Here full model seems to indicate the inclusion of all species/samples. However, “full model” also seems to be used to indicate a predictor variable (in e.g., Table 1). This is confusing.

134: It’s stated that Kipp’s index and divergence times are “also included” as predictor variables, but it needs to be clarified what original predictors they are being added to.

200: “Sx2” should be “S2”

237: Remove red highlight.

Some of the SI (e.g., Table S12) was not provided for review, so I am unable to comment on those materials.

Reviewer #2 (Remarks to the Author):

Review of the manuscript “Microevolutionary dynamics show tropical valleys are deeper for montane birds of the Atlantic Forest”

This study aims to test Janzen’s pivotal hypothesis “that mountains are higher in the tropics” at the population level in a community of birds in the Atlantic Forest of Brazil. Janzen’s hypothesis predicts higher genetic isolation and climate specialization in the tropics when compare with non-tropical habitats. Because the Atlantic forest is located both in the tropics and the subtropical region, a comparison between populations of the same species living inside and outside the tropics provide an excellent scenario to test these hypotheses. Authors obtain spatial and genetic data for the 21 species of birds sampled, making numerous comparisons between the tropical and non-tropical populations. Authors find that results support Janzen’s hypothesis at the population level: tropical populations show higher signals of isolation and climate specialization when compared with the non-tropical populations.

Disclosure: my field of expertise is phylogenetics and speciation, and I mostly work in at the macroevolutionary scale in the tropics. That being said I don't consider myself the best candidate to evaluate all the methods used by the authors.

Assessment:

This paper is one of the best examples I can think of a study that samples nature (vs. lab experiments) to test pivotal hypotheses regarding speciation in the tropical regions.

I praise the elegance and novelty of the study, the clarity of the writing, the thoughtful methods, and the appropriate interpretation of the results. This paper will be of wide interest in the community and will, without doubt, influence the field.

I believe this paper is ready to be published in its current form, and the only thing I would like to see added to the manuscript is a couple of phrases of how extinction can affect diversification differentially in tropical vs. non-tropical areas. For example, it has been proposed that Pleistocene climate oscillations affect differentially the two regions promoting pulses of high extinction followed by high speciation in non-tropical ecosystems (Weir and Schluter, 2007, The Latitudinal Gradient in Recent Speciation and Extinction Rates of Birds and Mammals, *Nature*: 315: 1574-1576). It is possible that the recent macroevolutionary studies that found higher diversification rates outside of the tropics are picking up the signals produced by the Pleistocene climatic fluctuations while the microevolutionary studies (i.g. the present study) do not pick up these signals.

It was a pleasure to review your study!

REVIEWER COMMENTS

Reviewer #1 (Remarks to the Author):

Thom et al. examine latitudinal differences in intraspecific population differentiation across space, focusing in on a latitudinal transect across the tropical/subtropical boundary in Brazil. They use well-sampled RAD-Seq datasets from 21 species to test for differences in the environments occupied, genetic structure, and dispersal between the subtropical and tropical regions. They find that species occupy lower elevations in the subtropics than the tropics. Tropical populations show greater differentiation across space and lower dispersal than subtropical counterparts. Tropical differentiation is associated with environmental turnover, whereas any subtropical differentiation is associated with distance alone.

Overall, I thought the conceptual focus on latitudinal differences in montane divergence was of fundamental interest, I appreciated the approach of examining the same species across a latitudinal transect, I thought the sampling was quite good, and I found the presentation to generally be effective. I have a few concerns regarding the data, I think the text could be clearer in places, and I thought some of the methods were lacking in detail or had redundant information provided across the Results, Methods, and Discussion. However, I think these issues could be corrected with a revision of the manuscript and would yield a very nice study of general interest.

We greatly appreciate all the comments and suggestions. We adjusted our manuscript acknowledging all points raised by the reviewer. Please find below a point-by-point response to all questions and comments.

Major comments:

1. Comparability of data among species and regions

Are ddRAD datasets comparable across species? If the same parameters are used for assembling datasets across species, those that contain more divergent alleles might be disproportionately affected by e.g., allele dropout. Might this impact any of the reported results? I know there is some prior work on these issues in RAD-Seq data (e.g., Ilut et al. 2014, Harvey et al. 2015) and ddRAD specifically (e.g., Cumer et al. 2021). I would like to see at least a discussion of the comparability issue.

This is a pertinent point, and we included a brief discussion on the new version of the manuscript. Please see *Material and Methods - Genomic Sampling*: “To minimize biases that can arise in comparative RADseq datasets due to variation in genetic diversity among species⁴⁷, we used relatively loose filtering for the clustering threshold (0.90 `clust_threshold`). This setting allowed for 10% sequence divergence between reads, which reduces the impact of allele dropout on species with more genetic diversity. Additional post-sequencing filters were not implemented because they have marginal impacts on estimates of genetic differentiation, admixture, and demographic parameters⁴⁸”

Of greater concern for this study would be a bias in diversity in the tropical versus subtropical regions. Greater sampling in the tropical region might lead to the discovery of more sites that are variable among individuals in that region. It does not look like more individuals were sampled in the Tropics based on e.g., Figs. 1 and 2, but I cannot be sure without access to Table S12. Greater sampling in one region would not bias genetic distances based on all sites in the data, but it could lead to upward bias in differentiation measures based on variant sites only. Was Nei's genetic distance based on all sites or variants only? Were all sites used for consStruct analyses or just SNPs? Please provide this information in the text.

To calculate Nei's genetic distance and estimate genetic structure with cosStruct we used SNPs only, thus genetic distance is relative rather than absolute. Information regarding the number of samples per region can be checked in Figure 2 (conStruct plot) and Figure S3. Table S12 was included on GitHub (<https://github.com/GregoryThom/>).

We included more detailed information regarding our sampling scheme in the discussion (see *Signal versus noise: the benefits of community-wide sampling*): "Sampling localities at different geographic scales or different numbers of samples between tropical and subtropical regions could also lead to biased results. For example, sampling at a closer geographic distance and within the same mountain in the subtropical region, but at distant localities and in different mountain ranges in the tropical region could by itself artificially produce lower genetic differentiation in pairwise genetic distance estimations in the former. However, we used a similar number of samples in both regions (Figure 2; Table S12). Additionally, our sampling scheme in the subtropical region is considerably more widespread than in the tropical region, covering distinct mountain ranges and an extensive geographical distribution (Figure S1, S2), suggesting that our results might be conservative in describing the higher degree of genetic differentiation within the tropical mountains."

2. PGLS presentation needs improvement

I understand the utility of the PGLS analysis to identify any drivers of additional demographic differences between the tropics and subtropics. However, I was confused why migration rates were not included as response variables, since the Fastsimcoal model included gene flow. I know the BSSM simulations already identified a difference in migration rates between the two regions, but would be curious to know if PGLS had the power to detect this. If not, it may indicate that PGLS using the parameters from Fastsimcoal lacks sufficient power to detect demographic differences between the regions. Also, the significant Tajima's D results needs explanation (see Minor Comments).

This is a great point. However, the fastsimcoal2 analyses did not estimate the amount of gene flow within regions, as the BSSM did, hence they are not comparable. The Fastsimcoal2 analyses estimated gene flow between tropical and subtropical populations, and given that the goal of PGLS analyses was to test differences within both regions, we did not include gene flow. Estimating gene flow between mountain regions was essential to properly estimate parameters of interest, such as effective population size, and population size changes, but it was not part of our main question.

We acknowledge the Tajima's D results on the discussion: "Interpreting the significant Tajima's D values was confounded by the greater population structure in tropical populations, which may have biased the metric towards finding more support for expansion."

I also found that methodological details were lacking for this PGLS analysis. What were the units of observation? Given that "region" was a predictor variable, each species must have been split into two so there would be separate values for tropical and subtropical populations, correct? This is not clear from the text (either the Results, the Methods, or the SI Methods).

Good point. We included a clearer description of all raised questions on the main body and removed redundant parts from SI (please see Material and Methods - *Testing for associations between the environment and genetic summary statistics*).

How were variables summarized within each population? Were segregating sites and pi calculated across all samples from those populations or were they calculated for each individual and a mean obtained?

Population genetics summary statistics were calculated per population, and not averaged per individual (please see Material and Methods - *Estimation of historical demographic parameters*).

How were distance metrics summarized – were they calculated as pairwise values for the subset of individuals in each subpopulation (tropical and subtropical)?

Distance matrices were generated for the entire distribution of the species (please see Material and Methods - Genetic structure between and within mountain regions), and pairwise comparisons between individuals occurring on the same mountain region were summarized by calculating an average (please see Materials and Methods - Testing for associations between the environment and genetic summary statistics).

The predictor variables are also unclear. Is region a simple binary variable?

Yes, it is a binary variable (Tropical or Subtropical)

How is TDiv calculated for each subpopulation given that there should be only one value for each species.

We estimated a single Tdiv per species using Fastsimcoal2. In our PGLS analyses, TDiv was duplicated on populations of the same species, and it was used to control for the time that the populations are occurring in the region.

To what does "Full model" refer – a multivariate analysis with region, Kipp's index, and TDiv all as predictors? This information could be added to the SI, which currently contains little more detail than the main text Materials and Methods.

Yes, it is a multivariate analysis with the region, Kipp's index, and TDiv. We adjusted Table 2 to clarify that.

The goal of the second PGLS analysis (testing level of conStruct structure versus environmental space) is less clear to me. I guess it was to identify whether the degree of environmental difference between the tropical and subtropical areas inhabited by a species determined whether it had a phylogeographic break between them?

This is perhaps not directly relevant to the major questions, so this analysis might be dropped. If it is retained, a sentence of justification should be added to the Results/Discussion section where the test is first introduced.

Also, in the methods, please describe the units of observation (I presume this was each species, unlike the first PGLS analysis) and how the difference between environments of the two regions was calculated.

Thanks for pointing this out. We agree that this PGLS analysis was not relevant for the main message of our study and we decided to remove it from the manuscript

3. Precision and clarity of text

I found the text unnecessarily wordy or vague in some places. For example, in the Abstract the sentence:

“However, distinguishing the background effects of species history and trait divergence from causal processes driving higher tropical diversity is challenging given the extent of evolutionary time separating taxa along latitudinal gradients.”

might be clarified as:

“However, distinguishing the impact of latitude versus phylogeny is challenging due to the evolutionary distance between tropical and temperate assemblages.”

It may sound drier, but it is more straightforward to interpret. There are other places where the wording can be tightened, some of which are indicated in the minor comments below.

We adjusted this sentence and reviewed the manuscript removing wordy and vague text.

4. Redundant methods details

The same methodological information is often duplicated (sometimes verbatim!) between the main text Materials and Methods, the SI Materials and Methods, and sometimes even in the Results and Discussion (see examples in Minor Comments). At the same time, key methods information is sometimes missing (see PGLS comment above). This meant I had to search multiple places to find key info and sort through the same details in each place, only to fail to find what I was looking for. Please try to reduce redundancy in favor of providing complementary detail in the SI.

Thanks for pointing out the redundant text. We removed duplicated method descriptions from the main body of the manuscript and SI.

Minor Comments:

49: "...and support that valley..." is awkward. Better: "...and support the [idea/theory] that valley...". There are several other places with the same issue (e.g., lines 129, 143).

Fixed

73: This idea would be clearer if you replaced "with contrasting" with "at very different".

Fixed

75: Replace "while" with "whereas".

Fixed

78: Based on these sentences, two knowledge gaps are being explored – (1) "continuous systems" containing close relatives and (2) intraspecific (rather than species-level) diversification. Perhaps make this clearer?

95: "Progressive" should be "progressively".

Fixed

116: Missing a word between "factors" and "driving" ("in"?).

Fixed

119: Remove comma after "configuration".

Fixed

125: "predict" should be "predicts".

Fixed

128: "on" should be "at".

Fixed

128: Which montane birds? The same 21 species used in genetic analyses?

The studied species. Fixed

169: What are "valley passes"? Just valleys?

We decided to use valley passes to match with the original term used by Janzen (1967; mountain passes). Here we use "valley" instead of "mountain" because what restricts individuals' movement are lower elevation areas.

182: It is misleading to say here that the best-fit model supported a significant effect of environmental distance. You could either present a pairwise comparison of otherwise-identical models with and without environmental distance, or you could say “the best-fit model contained environmental resistance variables”.

Fixed

183: “distance as significant” should be “distance as a significant”.

Fixed

190: My understanding is that dispersion refers to the pattern of distribution of individuals across the landscape, whereas dispersal refers to movement. It seems the two terms are used interchangeably here? I would suggest using dispersal when movement is involved.

Fixed

205: “individuals” should be “individual”.

We kept individuals

226: Was Tajima’s D lower in the subtropics? This might indicate population expansion and is worthy of discussion. Maybe there was just not enough power to identify differences in the population sizes from demographic modeling. The obtained values for Tajima’s D were relatively similar between mountain regions but were more negative for populations in the tropical mountains (Tajima’s D tropical: mean = -0.315, SD= 0.169; Tajima’s D subtropical: mean = -0.234, SD= 0.129). Although Tajima's D might be indicative of recent demographic changes, here we argue that the differences between tropical and subtropical mountains might be due to the distinct patterns of intra-population differentiation affecting pairwise differences between individuals. When explicitly estimating past population size changes with fastsimcoal2 we did not find any differences between mountain regions.

234: What was the purpose of this PGLS analysis? To determine what traits of species predicted whether there was a phylogeographic break between the two regions? Is this relevant? What does it mean that current environmental resistance is a significant predictor? Species with more environmental turnover among samples overall are more likely to have a phylogeographic break?

We included additional justification for this analysis in the results section (please see *Historical demography is not associated with mountain regions*).

240: This paragraph is odd in that it contains conclusions and then discusses sampling issues, which don’t really go together. I would suggest moving the conclusions sentences to the very end of the paper. Also, you might support the statements about sampling issues with quantitative information from the results or with additional sensitivity tests. This would also be a good place to include information addressing Major Comment 1.

We rephrase this section addressing the Major Comment 1

249: “taxa” is the plural of “taxon” (see also line 251).

Fixed

253: Is this accuracy demonstrated with simulations or similar? I could not find these details. If not, remove this sentence.

Yes, the accuracy of our model was assessed with cross-validations, where simulated data were used as pseudo-observed data, and estimated parameters were correlated with pseudo-observed parameters (see. Table S7). High correlation scores indicate the high accuracy of our model. A description of our cross-validation approach is available at Material and Methods: *Bidimensional stepping stone coalescent modeling*

261: Add “an” before “extensive”?

Fixed

267: Hyphenate “higher-latitude”.

Fixed

279: Could you more explicitly tie back in to Janzen’s hypothesis here? State explicitly if you confirm or refute it?

Fixed. We added “, confirming Janzen’s predictions within species”

283: This is a good discussion of the paradox of lower speciation but greater population differentiation in the tropics. My opinion is that the authors are correct that the likely explanation is a greater lag-time for differentiated populations to become species in the tropics. More work is certainly needed on this topic!

Thank you.

298: Insert “may be” between “which” and “cryptic”.

Fixed

303: “were” should be “was”.

We maintained as “were” given that we are referring to the 231 tissues

360: Clarify the normalization procedure. Was this a simple transformation?

Yes, we divided the values of each variable by the mean of the variable. We added, “Variables were normalized to the mean”.

650: State what method was used to obtain the barplots.

Fixed. We added “Effective dispersal in meters per generation for populations occurring in the tropical and subtropical mountains of the Atlantic Forest obtained with the Bidimensional Stepping Stone Model (BSSM)”

650: “spatial explicit” should be “spatially explicit”.

Fixed

668: Perhaps clarify that this regression was used to center the values across species?

These regressions were used to remove the effect of geographic distance from the other environmental matrices.

Table 1: Clarify that full model is the combined resistance layers (“full model” is not mentioned in the text).

Fixed

Table 2: What are the predictor variables (columns) here? Specifically, does region just refer to a binary variable or is this the environmental difference between regions as indicated in the SI methods? What is the “Full model”.

We added more info on the table’s legend and labels. The predictor variables were region, Kipp’s index, and divergence time between the two populations. Populations from the tropical and subtropical regions of the same species had the same value for divergence time. In the table, we show the t-value and p-value for each of these three predictors within the model (full model). Summary statistics and environmental matrices were the response variables.

Supplemental Information:

9: Check on table references (S5 does not contain sample information).

Fixed

111: This exact sentence appears in the main text. Please reduce redundant text.

Fixed

116: Almost this exact sentence appears in the main text. Please reduce redundant text.

Fixed

118: Here full model seems to indicate the inclusion of all species/samples. However, “full model” also seems to be used to indicate a predictor variable (in e.g., Table 1). This is confusing.

Fixed. This part was removed from the SI, and “full model” was replaced just by “model” on the main body.

134: It’s stated that Kipp’s index and divergence times are “also included” as predictor variables, but it needs to be clarified what original predictors they are being added to.

We added this sentence to the main text: “Here the normalized pairwise genetic distance between all individuals of all species was set as the response variable and pairwise environmental distances as predictor variables.”

200: “Sx2” should be “S2”

Fixed

237: Remove red highlight.

Fixed

Some of the SI (e.g., Table S12) was not provided for review, so I am unable to comment on those materials.

Tables and datasets are now available on <https://github.com/GregoryThom/>

Reviewer #2 (Remarks to the Author):

Review of the manuscript “Microevolutionary dynamics show tropical valleys are deeper for montane birds of the Atlantic Forest”

This study aims to test Janzen’s pivotal hypothesis “that mountains are higher in the tropics” at the population level in a community of birds in the Atlantic Forest of Brazil. Janzen’s hypothesis predicts higher genetic isolation and climate specialization in the tropics when compare with non-tropical habitats. Because the Atlantic forest is located both in the tropics and the subtropical region, a comparison between populations of the same species living inside and outside the tropics provide an excellent scenario to test these hypotheses. Authors obtain spatial and genetic data for the 21 species of birds sampled, making numerous comparisons between the tropical and non-tropical populations. Authors find that results support Janzen’s hypothesis at the population level: tropical populations show higher signals of isolation and climate specialization when compared with the non-tropical populations.

Disclosure: my field of expertise is phylogenetics and speciation, and I mostly work in at the macroevolutionary scale in the tropics. That being said I don’t consider myself the best candidate to evaluate all the methods used by the authors.

Assessment:

This paper is one of the best examples I can think of a study that samples nature (vs. lab experiments) to test pivotal hypotheses regarding speciation in the tropical regions.

I praise the elegance and novelty of the study, the clarity of the writing, the thoughtful methods, and the appropriate interpretation of the results. This paper will be of wide interest in the community and will, without doubt, influence the field.

We greatly appreciate the compliments on our manuscript and are very pleased to receive such positive feedback.

I believe this paper is ready to be published in its current form, and the only thing I would like to see added to the manuscript is a couple of phrases of how extinction can affect diversification differentially in tropical vs. non-tropical areas. For example, it has been proposed that Pleistocene climate oscillations affect differentially the two regions promoting pulses of high extinction followed by high speciation in non-tropical ecosystems (Weir and Schluter, 2007, The Latitudinal Gradient in Recent Speciation and Extinction Rates of Birds and Mammals, *Nature*: 315: 1574-1576). It is possible that the recent macroevolutionary studies that found higher diversification rates outside of the tropics are picking up the signals produced by the Pleistocene climatic fluctuations while the microevolutionary studies (i.g. the present study) do not pick up these signals.

We expanded our discussion to acknowledge the role of local extinctions on our data (please see *Historical demography is not associated with mountain regions*)

It was a pleasure to review your study!

Thank you!

REVIEWER COMMENTS

Reviewer #1 (Remarks to the Author):

Thom et al. have done an excellent job incorporating reviews from the first round, resulting in an improved paper and an excellent contribution to Nature Communications.

However, I do have one major concern that is raised by the authors' response to my first review and my re-reading of the paper. The authors imply in the conclusions that not only do tropical populations have higher levels of differentiation than subtropical, but also higher rates of differentiation. It's not clear to me whether this is supported by the results. Inferring a rate difference would require the inclusion of region-specific information on the time a species has been present. The authors indicate on line 171 that time since colonization of a region might impact differentiation, then indicate on line 173 that the minimum time a species has occurred in both regions is included as a covariable in the mixed-effect model. However, I believe the time used in this case was the same for both regions (the between-region divergence time or T_{div}). I believe the same was the case (T_{div} represents a single value for each species) in the PGLS analysis, based on the authors' response to my first review. Without region-specific values for time, I do not believe the authors can rule out the possibility that the populations distributed across tropical areas have not just been present for longer than the populations present in subtropical areas. My opinion is that this is the most likely explanation for the difference in differentiation between the regions, and this would be consistent with prior work reporting higher levels, but slower rates, of differentiation in tropical regions (and also older species, but slower speciation rates, in the tropics). I think the best solution to this issue is to remove the implication in the conclusions that rates of differentiation are higher in the tropics in this dataset.

Otherwise, the paper looks good and I have only a few minor comments (by line number):

106: "north to" should be "north of" and "south to" should be "south of"

174: Specify that the divergence time between populations was from the fastsimcoal 2-population model (if correct).

192: Define "Nm" at first use in the paper.

Reviewer #2 (Remarks to the Author):

As stated in my initial assessment of the manuscript, I believe “Microevolutionary dynamics show tropical valleys are deeper for montane birds of the Atlantic Forest” is a great contribution to the field and will be of interest for a broad audience.

The current version of the manuscript addressed all my previous comments and I believe the paper is ready to be published.

Minor comments:

line 253: replace "moure" with "more"?

Reviewer #4 (Remarks to the Author):

This is a great manuscript, with a very solid set of analyses and a sound interpretation of the results, in my view. However, I found many parts of the text confusing and sometimes even misleading, and have made suggestions to improve this situation. Clearly, the text needs to be polished up for clarity and sometimes sentences have to be connected in a more logical way. I have made rather extensive and detailed suggestions in this regard as comments directly onto the ms. PDF. I hope the authors find them useful and also that they somehow contribute to improve even more their excellent work.

Reviewer #1 (Remarks to the Author):

Thom et al. have done an excellent job incorporating reviews from the first round, resulting in an improved paper and an excellent contribution to Nature Communications.

Thank you. We adjusted our manuscript acknowledging all points raised in your comments.

However, I do have one major concern that is raised by the authors' response to my first review and my re-reading of the paper. The authors imply in the conclusions that not only do tropical populations have higher levels of differentiation than subtropical, but also higher rates of differentiation. It's not clear to me whether this is supported by the results. Inferring a rate difference would require the inclusion of region-specific information on the time a species has been present. The authors indicate on line 171 that time since colonization of a region might impact differentiation, then indicate on line 173 that the minimum time a species has occurred in both regions is included as a covariable in the mixed-effect model. However, I believe the time used in this case was the same for both regions (the between-region divergence time or T_{div}). I believe the same was the case (T_{div} represents a single value for each species) in the PGLS analysis, based on the authors' response to my first review. Without region-specific values for time, I do not believe the authors can rule out the possibility that the populations distributed across tropical areas have not just been present for longer than the populations present in subtropical areas. My opinion is that this is the most likely explanation for the difference in differentiation between the regions, and this would be consistent with prior work reporting higher levels, but slower rates, of differentiation in tropical regions (and also older species, but slower speciation rates, in the tropics). I think the best solution to this issue is to remove the implication in the conclusions that rates of differentiation are higher in the tropics in this dataset.

Thanks for the comment. We decided to keep this section. We rephrase the paragraph by adding potential caveats and justifications of why we think populations of all species are unlikely to be older in the tropics. We added: "However, we did not directly estimate rates of differentiation, and not accounting for time in the landscape, beyond population split times, could bias interpretations. At broader geographic scales, the degree of phylogeographic structuring in birds was previously shown to be best predicted by species age¹⁵, which applied to this system could indicate tropical populations have greater structure, in part, because they have been in lower latitudes longer. Despite the possibility of differential time in the landscape, genetic differentiation within the Atlantic Forest tropical mountains was likely formed after the divergence between most tropical and subtropical populations, following the upslope movement of montane environments associated with past climatic cycles³⁹. Future studies that model the deeper biogeographic history of these montane taxa will be able to clarify the role of time-dependency in interpreting rates of genetic structuring across the latitudinal gradient."

Otherwise, the paper looks good and I have only a few minor comments (by line number):

106: "north to" should be "north of" and "south to" should be "south of"

Fixed

174: Specify that the divergence time between populations was from the fastsimcoal 2-population model (if correct).

Yes, this is correct. Fixed

192: Define "Nm" at first use in the paper.

Fixed. We added "Nm is the product of the effective population number and rate of migration among populations"

Reviewer #2 (Remarks to the Author):

As stated in my initial assessment of the manuscript, I believe "Microevolutionary dynamics show tropical valleys are deeper for montane birds of the Atlantic Forest" is a great contribution to the field and will be of interest for a broad audience.

The current version of the manuscript addressed all my previous comments and I believe the paper is ready to be published.

Minor comments:

line 253: replace "moure" with "more"?

Fixed

Reviewer #4 (Remarks to the Author):

This is a great manuscript, with a very solid set of analyses and a sound interpretation of the results, in my view. However, I found many parts of the text confusing and sometimes even misleading, and have made suggestions to improve this situation. Clearly, the text needs to be polished up for clarity and sometimes sentences have to be connected in a more logical way. I have made rather extensive and detailed suggestions in this regard as comments directly onto the ms. PDF. I hope the authors find them useful and also that they somehow contribute to improve even more their excellent work.

We greatly appreciate all the comments and suggestions that greatly improved the quality of our manuscript. We adjusted our manuscript acknowledging all points raised by the reviewer. Please find below a point-by-point response to all questions and comments.

Line 71: What's Janzen's hypothesis after all? Ok, "gradients have greater thermal overlap in temperate regions", but how does that actually translate into a hypothesis. You guys have not made that clear, so the jump to how it acts "between populations within species" cannot be made without a more elaborate discussion on the implications and even predictions of Janzen's hypothesis.

We addressed those topics in the second and third paragraphs of the introduction, where we state predictions and provide a set of implications for Janzen's hypothesis.

Line 72: Again, the link here between "initial stages of diversification" and Janzen's hypothesis is not clear. We addressed those topics in the second and third paragraphs of the introduction

Line 84: To be clearer: state a clear correlation instance: "populations should occupy wider elevational ranges and broader environmental space AS LATITUDE INCREASES (or decreases - this may not be the case if I understood the hypothesis correctly)". But I hope you guys got the idea.

Thanks for the suggestion. We added a correlation between processes.

Line 87: Why not replace "increasing climate variability" with just climate or climate niches. Cold or hot. Dry or humid. The sentence immediately below that links to general climate patterns, not variability. Remember, both tropical / subtropical or low/latitude climates can be very variable, so you are really not interested in variability here, but just on general patterns. Things like this render the text less fluid.

We replaced "increasing climate variability" by "regional differences in climate"

Line 91: OK, but the critical mechanism here that seems missing to me is that gene flow amongst low elevation / high latitude populations would be higher than among high elevation / low latitude (tropical) populations. In other words, the area (climate niches) occupied by low elevation / high latitude populations is greater than those inhabited by high elevation / low latitude (tropical) populations, so in addition to more connectivity and gene flow among low elevation / high latitude, you could also have greater effective pop. sizes, which also implicate in higher genetic similarity. Again, I feel the need to explain more clearly the logical reasoning and the theoretical underpinnings behind these expectations.

We added "For instance, we expect higher rates of gene flow and larger effective population sizes for the low elevation, high latitude populations." to the text.

Line 94: I would not give up still focusing at the microevolutionary level here. For instance, the better alternative to be discussed here would be isolation by distance, and then you could still keep comparisons and null / alternative hypotheses at the same microevolutionary level.

We understand the point and agree that an alternative scenario for lack of differences regarding population dynamics associated with climate would be IBD, but the main hypothesis we tested are based on macroevolutionary patterns that are observable within species along a narrow latitudinal gradient. Thus an alternative explanation for lack of differences on population dynamics between tropical and subtropical populations would be that the differences that were predicted by Janzen's model are only observable when exploring large latitudinal gradients and highly divergent species.

Line 103: To me, "separated" here would be a more simple term to reflect that there is gap between these two formations.

Fixed

Line 135: Mention climate here to be more explicit, by adding in parenthesis: "(annual temperature range)".

Fixed

Line 223: There might be more room here for exploring or just mentioning the alternative view (which you dismiss here), i.e., that Tajima's D could be showing that tropical populations experienced more drastic population changes than the subtropical ones. Because Tajima's statistic reflects a departure from the mutation - drift, equilibrium, it may indicate bottlenecks as well, not necessarily expansion. Look Figs. 3c and 3d. The gap separating tropical and subtropical populations is bigger when mean stability resistance is considered, in comparison to environmental resistance. So, historically, landscape resistance was higher (slightly?) for tropical populations and lower for subtropical populations, which could implicate in higher demographic instability of tropical populations, when compared to the subtropical ones, thus in accordance with the Tajima's results. By the way, I find the term "Mean stability resistance" a little bit misleading, and would prefer it to be replaced with "mean historical environmental resistance", or just "mean historical resistance".

Thank you for the comment. For both regions Tajima's D had negative values, meaning that there are fewer haplotypes than segregating sites. This pattern is expected to be observed in recent selective sweeps or population demographic expansions after a recent bottleneck. Nevertheless, values were relatively close to zero. Here we argue that the more negative values of Tajima's D in the tropical populations (stronger signal for expansion) might have been affected by the higher restriction to gene flow between individuals. Although we found higher historical resistance in the tropical mountains our demographic parameter estimation approach and summary statistics do not support bottlenecks for tropical populations. Higher historical resistance in tropical mountains does not necessarily imply more

intense demographic size changes, given that slight changes in the elevational distribution of high elevation habitats can drastically affect individuals connectivity without changing substantially the occupied area. Figures 3c and 3d are not necessarily comparable, given that in figure 3d we removed the effects of geographic distance from the current environmental resistance. We changed "Mean stability resistance" to "mean historical environmental resistance" as requested.

Line 391: This is very loose in the text here...is this because there is more connectivity among subtropical populations, which inhabit areas with higher annual temperature variation? Throughout the manuscript, I find hard to grasp the authors's concept of seasonality. I have asked above for them to specify which type of seasonality they are talking about whenever possible. For example, is Bio4 derived from annual temperature variation? If yes, then I believe they have use their concept of seasonality consistently throughout the text, and always making clear that they are talking about annual variation, and not, for instance, rainfall, etc.

Thanks for pointing this out. We reviewed the text to make it clear that we are talking about temperature seasonality

Line 393: I miss here some minimum detail about the historical time frame being compared, so I would try to squiz in here a short fragment of the text in the SI materials referring to the historical time frame considered: "1) Pleistocene: Last Interglacial (ca. 130 ka), 2) Pleistocene: Last Glacial Maximum (ca. 21 ka), 3) mid-Holocene: Northgrippian (8.326-4.2 ka), and 4) late-Holocene: Meghalayan (4.2-0.3 ka)."

Fixed

Line 703: Replace North and South in the X-axis with Tropical and Subtropical, respectively.

Fixed

REVIEWERS' COMMENTS

Reviewer #1 (Remarks to the Author):

Thom et al. have sufficiently addressed the additional concerns I had in the second round of review. I think this is an excellent contribution and look forward to seeing it in print.

Reviewer #4 (Remarks to the Author):

I am happy with all the changes and with the way the authors have addressed the comments made by myself and other reviewers. So, I am happy to recommending it for publication.